# Spite is contagious in dynamic networks

Zachary Fulker [1], Patrick Forber [2], Rory Smead [3] & Christoph Riedl [1,4,5,6 ✉]

Spite, costly behavior that harms others, presents an evolutionary puzzle: given that both the actor and recipient do worse, how could it emerge? We show that dynamically evolving interaction networks provide a novel explanation for the evolution of costly harm. Previous work has shown that anti-correlated interaction (e.g., negative assortment or negative relatedness) among behavioral strategies in populations can lead to the evolution of costly harm. We show that these approaches are blind to important features of interaction brought about by a co-evolution of network and behavior and that these features enable the emergence of spite. We analyze a new model in which agents can inflict harm on others at a cost to themselves, and simultaneously learn how to behave and with whom to interact. We find spite emerges reliably under a wide range of conditions. Our model reveals that when interactions occur in dynamic networks the population can exhibit correlated and anti-correlated behavioral interactions simultaneously, something not possible in standard models. In dynamic networks spite evolves due to transient and partial anti-correlated interaction, even when other behaviors are positively correlated and average degree of correlated interaction in the population is low.

[1] Network Science Institute, Northeastern University, Boston, MA, USA. [2] Department of Philosophy, Tufts University, Medford, MA, USA. [3] Department of Philosophy and Religion, Northeastern University, Boston, MA, USA. [4] D'Amore-McKim School of Business, Northeastern University, Boston, MA, USA. [5] Khoury College of Computer Sciences, Northeastern University, Boston, MA, USA. [6] IMT Lucca, Piazza S. Ponziano, 6, 55100 Lucca, Italy. ✉email: c.riedl@neu.edu

Costly behavior that harms others, sometimes known as spite[1–3], is among the most basic of anti-social behaviors. However, it presents an evolutionary puzzle. If there are no benefits, how could it have emerged? There nevertheless are cases of costly harm in both humans[4–6] and non-humans[7–10]. For instance, in human evolution there is a case to be made that the killing of individuals who exhibit high degrees of reactive aggression, clearly a dangerous harmful endeavor, played an important role in the self-domestication of human ancestors[11]. This has prompted a number of evolutionary models to explain the emergence of such costly harm. We can represent this sort of interaction abstractly with a simple game. Suppose two individuals interact and each has the opportunity to pay a cost to inflict a harm on the other. Let $b$ be the benefit derived from normal social interaction and let $-c$ be the cost an agent can pay to take away that baseline benefit. This game is the Prisoner's Delight[12], which contrasts with the Prisoner's Dilemma in that acting anti-socially, rather than pro-socially, is costly (Fig. 1). In this game, unlike in the classic dilemma, harmful behavior is strictly dominated and generally we should not expect it to evolve.

Standard models assume large randomly mixing populations and, in such models, spiteful behavior is invariably eliminated. Allowing non-random interaction due to assortment of strategies opens new possibilities. It is well known that costly altruism can evolve with non-random interactions. In particular, altruism can evolve if interactions are correlated—that is, if altruists interact disproportionately with other altruists[13–15]. A similar, but inverse, result has been shown regarding spite[16]. If a given strategy or type interacts disproportionately often with different types—if interactions are anti-correlated—spite can emerge by generating relative advantages. To illustrate this, consider a population where individuals can play one of two possible strategies: 'Social' or 'Spiteful'. 'Spiteful' agents choose to pay a cost to inflict harm on their interaction partner, while 'Social' agents do not pay a cost and do not inflict harm. Note that the strategy label 'Social' is used minimally to refer to non-spiteful behavior. Let $r$ denote the degree of anti-correlated behavior in the population: with probability $r$ individuals interact with those using different strategies and with probability $(1 - r)$ interact randomly. Using the payoffs from Fig. 1, we can deduce that 'Spiteful' behavior will outperform 'Social' behavior whenever the degree of anti-correlated interaction is larger than the cost-to-harm ratio: $r > c/b$ (see SI). This rule mirrors Hamilton's rule for when altruism will be favored, and was also derived by Hamilton in the context of negative relatedness[16,17]. However, anti-correlated interactions need not be generated by genetic relatedness. It can be realized in any number of ways including green beard effects[8], neutral markers[18], spatial structure[19], and small populations[20]. The exogenous parameter $r$ represents correlated interactions abstractly, without reference to a specific mechanism. We demonstrate that dynamic networks can endogenously produce the anti-correlated interactions necessary for spite, and that dynamic networks do so in a way that is not captured with the traditional analysis.

Dynamic networks capture the fact that in many human interactions we choose the people we wish to interact with—and how often—such as when we join social or religious groups[21]. It has been shown that dynamic networks allow for endogenous correlated interactions and thus for the emergence of altruism in the classic Prisoner's Dilemma[22–26]. We show that dynamic networks can also do the opposite: produce anti-correlated interactions which allow for the spread of spite. Specifically, spite will spread in dynamic networks with adaptive link weights, where these weights are based on reinforcement learning. This happens because both correlated and anti-correlated interactions occur simultaneously, enabling spite to spread through the network via imitation. Reinforcement learning has been identified by cognitive scientists and economists for its ability to describe human choice in a variety of iterative decision-making settings that is biologically plausible[27–29]. We use reinforcement learning[22,30,31] to dynamically update network ties, and find that correlated and anti-correlated interactions emerge endogenously in a population, which impacts the evolution of cooperation and social conventions. Our approach contrasts with the class of dynamic network models that represent network links as discrete and where the dynamics describe patterns of link breakage and

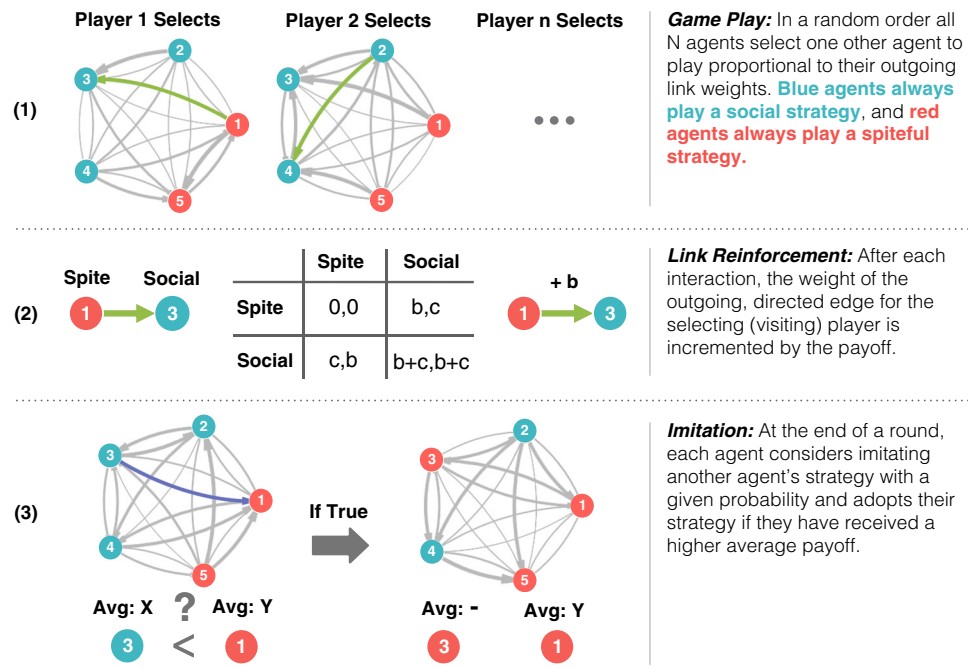

**Fig. 1 Game play and updating mechanism.** In each round, agents must (**1**) select an interaction partner, (**2**) update their network weight based on the payoff, and (**3**) consider imitation. We normalize the payoffs by setting $b + c = 1$ and examine variation in the ratio $b/c$.

formation[24,26,32]. Real social network ties are rarely discrete, therefore modeling network connections with reinforcement learning adds an important element of realism[33–35].

Here we employ these weighted dynamic networks to study the evolution of costly harmful behavior in the Prisoner's Delight. We track the emergent levels of correlated and anti-correlated interactions among the strategies over time and allow the strategies to co-evolve over time via imitation. 'Social' agents become correlated with one another and 'Spiteful' agents become anti-correlated. Despite the advantages of pro-social behavior in this game and the ability to correlate social interactions, spite nevertheless evolves in a wide range of conditions.

## Results

We use agent-based computer simulations[36] to explore the co-evolution of network structure and behavioral strategies. Results are averaged across 200 randomly seeded simulations with 1 million time-steps and a population of 50 with an imitation rate of 0.01, unless otherwise noted. Each time-step every agent selects one other agent to visit and both agents play the game. Agents then update their likelihood to visit others based on the payoffs received and have a chance to imitate the strategies other agents (Fig. 1). Because playing 'Social' is a dominant strategy in the Prisoner's Delight, it will be expected to fixate in a randomly mixing population. Any agent playing 'Spiteful' in a given interaction will be worse off than if they had played 'Social' instead. However, when $b > c$, 'Spiteful' agents inflict greater harm on their partner than they pay in cost. This difference allows for the possibility of the 'Spiteful' strategy to spread via imitation in non-random networks. Setting $b + c = 1$ normalizes the payoffs and allows us to analyze the effects of payoff differences through the ratio $b/c$.

**Spite spreads on dynamic networks**. Our key question is whether reinforcement learning on a dynamic network can produce anti-correlated interactions leading to the emergence of spiteful behavior: the answer is yes. All populations converge to a uniformity of social behavior or spiteful behavior. Spite becomes the norm in our model over a range of population sizes (Fig. 2a), as well as imitation rates (Fig. 2b). This outcome occurs when the ratio of the harm done to the cost of spite is sufficiently large, i.e., a high enough $b/c$ value. As each agent reinforces their network weights based on the payoffs they receive, over time both 'Social' and 'Spiteful' agents learn to visit 'Social' agents. This learning process changes the share of all interactions that each of the four possible ordered visitor–host couplings represents (Fig. 2c). The result is that 'Spiteful' agents begin to participate in fewer interactions as other agents learn to avoid visiting them. In each round, however, every agent has their own opportunity to select a partner to visit. It becomes likely that a 'Spiteful' agent will select a 'Social' agent to visit, and therefore receive a payoff of $b$. Conversely, 'Social' agents are visited more frequently as other agents learn to target them. These interactions will occur with both 'Social' and 'Spiteful' agents, providing a mixture of the maximum payoff, 1, and numerous much smaller payoffs, $c$. As the ratio of $b/c$ grows larger, the smaller value of $c$ will slow the rate of growth in the average payoff received by 'Social' agents. At the same time, the increased value of $b$ and decreasing visits from other 'Spiteful' agents will increase average payoff received by 'Spiteful' agents until it surpasses that of the 'Social' agents (Fig. 2d). This means when 'Social' agents select a 'Spiteful' agent to consider imitating, they will choose to adopt the 'Spiteful' strategy.

These central results are robust with respect to starting population frequencies: spite is able to spread reliably to the whole population from a single 'Spiteful' individual (Fig. S2). Similar results are also derivable from a simplified analytic model assuming discrete ties formed by choosing best-responses (see 'Methods'). In the simplified model, a single 'Spiteful' individual will invade and spread through the whole population provided $3b - c > 2$, even if overall average interactions are positively correlated when invasion occurs. Just as in our simulation results the invasion of 'Spiteful' occurs because 'Social' agents make up a larger proportion of the 'Spiteful' agents' interactions than they do for other 'Social' agents. This is possible because 'Social' agents gain incoming 'Spiteful' links, reducing their share of 'Social' interactions. Simultaneously, 'Spiteful' agents shed harmful links with other 'Spiteful' agents, increasing their share of 'Social' interactions.

The endogenous partner choices of the agents in response to payoffs is essential. This allows 'Spiteful' agents to disproportionately target 'Social' agents, even as the 'Social' agents are disproportionately interacting with one another. The simultaneous independent formation of anti-correlated interactions for 'Spiteful' agents as well as correlated interactions for 'Social' agents is made possible by the use of asymmetric link updating in the model. This allows each agent to form reinforced preferences over the other agents they can choose to visit without being influenced by who approaches them. This would not be possible under symmetric link updating: then one agent's preference for frequently visiting another agent would cause the visited agent to add reinforcing link weight to the visiting agent. Since both kinds of correlated interaction can appear simultaneously, we represent the mean degree of correlated interaction as $\bar{a}$. If $\bar{a} > 0$, mean interactions are positively correlated. If $\bar{a} < 0$, mean interactions are anti-correlated. The classic condition for the invasion and stability of spite in a population corresponds to an overall degree of correlated interaction below $-c/b$ (see SI). Note that although we can compare the results, $\bar{a}$ is distinct from the classic exogenous $r$-parameter as $\bar{a}$ is merely a descriptive statistic of the network structure resulting from the local learning process of agents. Agents gradually adapt their behavior based on past payoffs and everyone learns to avoid 'Spiteful' agents. This tends to produce correlated interactions among agents engaged in 'Social' interactions, and anti-correlated interactions between agents engaged in 'Spiteful' interactions (Fig. 2e). This outcome highlights a frequently overlooked detail of previous modeling approaches: it is important to consider and measure the degree of correlated interaction for each strategy type individually, in addition to the overall value. In our model, the overall average degree of correlated interaction stays relatively stable and neutral over time because the offsetting correlated interactions of each strategy type. A notable consequence of this is that spite can reliably spread through a population even with relatively neutral correlated interaction ($\bar{a} > -c/b$; Fig. 2f). This result suggests a significant limitation of the exogenous global methods of modeling correlated interactions and that the well established $r > c/b$ condition for the evolution of harmful behavior does not generalize.

**Effect of variations in learning**. Network learning speed influences the evolution of network weights and the resulting degree of correlated interaction. The rate of learning can be adjusted by multiplying the payoffs received in the reinforcement process. Stronger adjustments of the relative weight of agents' outgoing network weights speed up the process of finding partners that they perform better against. Thus, faster network learning causes the correlated and anti-correlated interactions of 'Social' and 'Spiteful' agents to emerge earlier, leading to the 'Spiteful' strategy to become dominant after fewer time steps. Given a

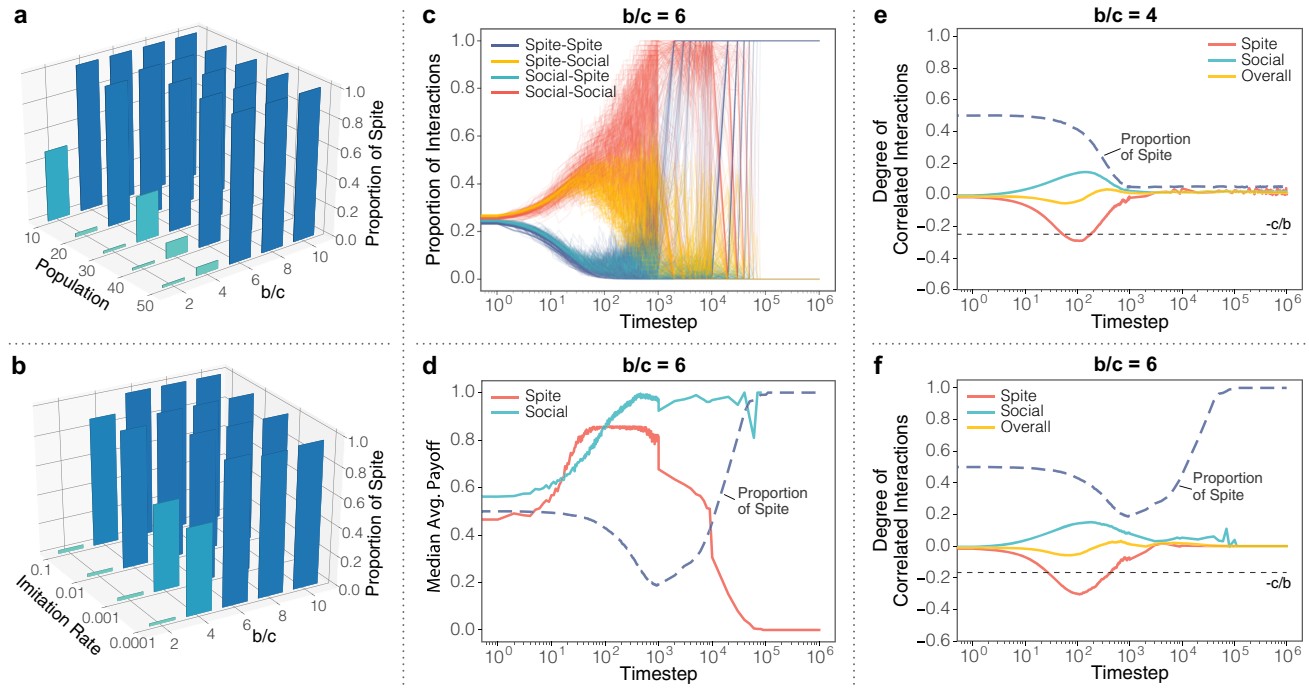

**Fig. 2 Dynamic networks produce (anti-)correlated interactions that lead to spite. a** 'Spiteful' fixates at the final time-step of our model over a range of population sizes, and **b** imitation rates. **c** The local network learning process of each agent changes the likelihood of each of the possible pairwise interactions over time (plotted by seed). 'Spiteful' agents learn to target 'Social' agents, who provide the highest payoff, and not each other. Similarly, but inversely, 'Social' agents learn to target each other, and to avoid 'Spiteful' agents. **d** The network learning process significantly improves the median average payoff of 'Spiteful' agents (**d**–**f** averages over 200 simulations). **e** The system level result of these local learning processes is the formation of positively correlated interaction patterns among 'Social' agents, anti-correlated interaction patterns among 'Spiteful' agents, and relatively neutral overall correlated interactions. **f** Under the interaction patterns emerging in the model, if the ratio $b$ to $c$ is sufficiently large, then spite emerges. Interestingly, the standard condition for the invasion and stability of spite, overall correlated interactions below $-c/b$, does not need to be satisfied for the 'Spiteful' strategy to emerge in our model.

non-negligible network learning speed and a sufficient number of time steps, however, spite regularly emerges in the population provided a large enough $b/c$ value (Fig. 3a). Network learning speed also effects the rate at which global network patterns appear and transition throughout the simulation. The overall network displays four key stages during the network evolution process (Fig. 3b). Populations begin with behavior akin to random mixing, due to the fact that agents are initialized with uniform network weights. Structure in the population begins to emerge as agents learn to target 'Social' agents in their interactions. 'Social' agents begin to receive most incoming interactions from both 'Social' agents and 'Spiteful' agents. This process continues and results in a core–periphery structure with 'Social' agents forming the core, and 'Spiteful' agents generally in the periphery receiving relatively few visitors. Despite this, the 'Spiteful' strategy begins to spread by imitation due to comparison of relative payoffs between individuals. Finally, the 'Spiteful' strategy comes to dominate and the network slowly trends back toward random mixing as there are no more 'Social' agents to target.

Another parameter that influences network evolution is the discount rate in learning, which controls the rate at which weights diminish over time. Network discounting determines how significant recent payoffs are relative to total past payoffs in determining network weights. This can be interpreted as an abstract representation of recency bias or fading memory of past payoffs. This is important both as a psychologically realistic aspect of learning and since discounting is known to impact results of reinforcement learning rules[22,27–29].

Network discounting allows agents to unlearn network link weights connected to agents who have switched from the 'Social'

to 'Spiteful' strategy. This is implemented at the end of every round by multiplying the weight of each network connection by a constant $(1 - \delta)$ where $\delta$ $(0 \leq \delta \leq 1)$ represents the degree to which past payoffs are discounted. When $\delta \approx 0$, learning speed slows down over time and agents will not adapt to changes in the strategies of other players. If a frequently visited agent changes strategy from 'Social' to 'Spiteful', the lack of new reinforcement combined with the discount factor allows agents to unlearn their previous reinforced behavior. In practice, this means that when the network discount is small, it dampens the levels of correlation and anti-correlation in the model (Fig. 3c). When the network discount value $(1 - \delta)$ is set to 0.01, or larger, spite emerges over a large space of parameter combinations. But when the network discount is set to 0.001, spite does not become the dominate strategy even for very high values of the network learning speed.

Thus far, we have only considered one imitation mechanism, but we can easily implement others in the model. For example, if the mechanism imitated behaviors based on total payoffs rather than average payoff per interaction, spite rapidly dies out and populations always converge on the 'Social' strategy. The opposite is true of cooperation in the Prisoner's Dilemma[14,22] (see Supplementary Note 4 for comparison). Furthermore, we can implement alternative imitation mechanisms that mirror biological reproduction, copying other strategies with a probability proportional to their success such as the Moran process[37,38]. Adapting the Moran process to our model shows that spite can still emerge with considerable frequency, albeit with less regularity (see Supplementary Note 6 for more detail).

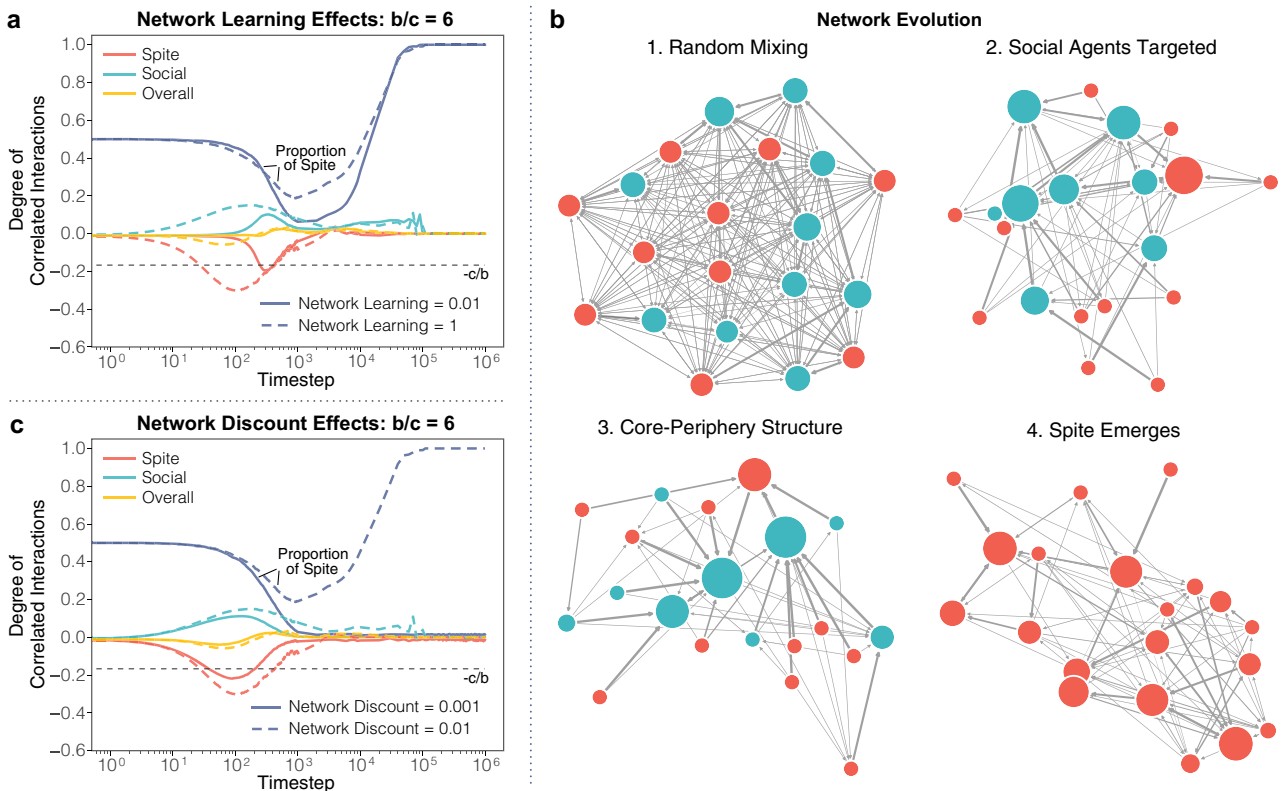

**Fig. 3 Network processes drive model results. a** Even when network learning is extremely slow, spite can still emerge in the population. The formation of correlated interactions is delayed when compared to our standard learning speed of 1 (dashed lines), but the end result is the same. **b** Stages of network evolution. The system is initialized in a state of random mixing, and as the learning process occurs, 'Social' agents gain incoming link weight and 'Spiteful' agents lose incoming link weight. This process transitions network structure to a tightly connected core of highly-target 'Social' agents and a periphery primarily made up of 'Spiteful' agents. Finally, once spite emerges in the model, network discounting causes the system to trend back toward random mixing. **c** Very small values of network discounting prevent the emergence of spite by slowing how quickly agents can forget their connections to 'Spiteful' agents. This process reduces the magnitude of correlations when compared to the standard model with a discounting rate of 0.01.

## Discussion

Our model shows that costly harm can spread through a population, structured by preferential interaction, via imitation. Such harm can emerge even without a significant overall degree of anti-correlated interactions in the population. It is well known that in populations with a sufficient degree of anti-correlated interaction that costly harm will emerge[8,12,16,18–20]. What our model demonstrates is that the degree of anti-correlated interaction can be partial, transient, strategy specific, and that it can coexist with correlated interactions among other strategies. These results highlight the importance of evaluating patterns of correlated interaction not just at a system level, but also at the level of strategy type. Consequently, traditional global-parameter methods of representing such assortment in populations cannot adequately capture the evolutionary consequences of dynamic assortment on evolving networks. Thus, our results reveal a novel evolutionary pathway for the emergence of costly harmful behavior in any system where individual agents are capable of learning by reinforcement and imitation. It has been recognized that dynamic networks can provide a mechanism for the spread of cooperation[22–26], and our results reveal they can also cause the spread of spite. Interestingly, recent empirical work shows that among humans, anti-social behaviors spread more readily among peers than do pro-social behaviors[39].

There are two connections between our model and other formal evolutionary models of social behavior that help clarify the generality of our results. First, while we have constructed our model using learning dynamics, the abstract nature of learning models allow for a more general application. Learning models tend to be associated with within-generation cultural evolutionary interpretations[40]. Yet there are formal results that show that models of imitation learning map onto biological reproduction[41,42], and models of reinforcement learning map onto standard Darwinian evolutionary dynamics[43]. This allows for biological interpretations of the model in addition to the cultural interpretations. Second, while we have framed our research in terms of correlated interaction, there are many important formal results connecting the notion of correlation to the notions of relatedness and inclusive fitness in Hamilton's work[8,15,17,44,45]. We find that our model has similar results when using a strategy update rule more amenable a biological interpretation (specifically a modified Moran process; see SI). This further supports the view that the overall effect we observe should be relevant to any species that is capable of preferential interaction. A biological interpretation of our model would then have implications for whether we need to re-evaluate the traditional global-parameter approaches to kin-selection and inclusive fitness.

Our results also have significant implications for open questions in social science and the evolution of social behavior. Human (and human ancestor) populations are clearly candidates for the operation of this evolutionary mechanism. Humans often choose their interaction partners on the basis of past experience and imitate the social behavior of others. Humans are also tuned to the success of their behavioral strategies and can update their behavior accordingly. That said, our model focuses only on the

costs and benefits of behavioral strategies and does not include motivational states, sophisticated cognition, or many other psychological mechanisms. A full account of human social behavior would require considering these factors explicitly. Applying these models to humans, or any cognitively complex organism, raises a host of questions about how psychological mechanisms relate to behavior, e.g., about how motivations connect to evolutionary models of behavioral change. One avenue for further research is to pursue the connections between formal models, such as ours, and comparative psychological studies on altruism and spite[4,46–48].

The origin and evolution of punishment is another area of ongoing debate and research where our results are relevant. The use of punishment has been well-document in humans and broader ecological systems[49–51]. Punishment, broadly construed, is costly harm inflicted conditional on some behavioral response. An example of such behavior occurs in laboratory experiments of iterative public goods games in labor settings with profit sharing[52]. In these settings, humans choose to reduce the payoff of a non-contributing free-rider even when they are charged a cost in order to do so[53]. Punishment can influence social interaction in many possible ways. Enforcing norms through punishment can stabilize cooperation (or any other behavior)[54], or create complicated interactions between reciprocity and retaliation[55,56]. There are also a number of hypotheses on the evolution of punishment[57]. One influential approach treats punishment as an altruistic behavior that stabilizes cooperative norms[58]. More recent studies have shown surprising complexity to punishing behavior, including anti-social punishment[6,59], where individuals punish cooperating members and can destabilize cooperation[60]. Our study, by uncovering a new pathway for the evolution of costly harm, suggests a new hypothesis for the origin of punishment. Rather than evolving to stabilize a cooperative social environment by enforcing some beneficial norm, costly harm may have emerged independently as a way of targeting competitors. Then it need only be directed toward enforcement of some other behavior to become true punishment[3,61].

## Methods
We model the co-evolution of interaction structure and social behavior[62,63]. Agents are assigned initial behaviors in the Prisoner's Delight game: 'Social' or 'Spiteful'. Agents choose their interaction partner, update who they are likely to choose on the basis of payoffs received, and periodically imitate others who are receiving higher average payoffs.

**Simulation model**. Network updating occurs via Roth-Erev reinforcement learning. Each round, agents select one other agent to visit proportional to their outgoing network weights, interact, and receive a payoff. Network weights are updated according to Roth-Erev reinforcement learning on the basis of the payoff received: the larger the payoff, the more weight is added to that agents out-going network link. Network weights are updated asymmetrically, meaning only the agent who initiated the interaction (the visitor) updates their outgoing weight based on the received payoff. This asymmetry represents a situation where individuals can control who they choose to approach, but not who approaches them. It also allows some individuals, who receive more visitors than others, to have more interactions per round (however, each agent is guaranteed one interaction as visitor). Our network model also represents links in a continuous manner. This approach avoids the need to exogenously set the number of network connections which can affect results[64]. Instead, all learning and evolution of network weights occurs endogenously. Network weights are initialized as uniform, representing random initial interaction. Structured interaction emerges as agents learn and payoffs accumulate.

Strategy learning occurs via imitation. After each round is complete, agents have a chance to consider imitating the strategy of another agent. When imitating, an agent selects another agent with a probability proportional to how often they interact, and if the selected agent received a higher average payoff per interaction in the previous round, imitates that agent's strategy. This captures the importance of social learning in many animal behaviors, while also allowing individuals to select their interactions based on their own experiences[22,65,66]. Imitation is a form of social learning that is used in a variety of contexts to model cultural evolution[40,67] and also has important formal relationships to models of biological evolution[41]. Initial strategies are determined by randomly assigning half of the agents to be

'Social' and the other half to be 'Spiteful' (Supplement Note 5 also examines the case of a single random 'Spiteful' agent).

More precisely, we model a set of $N$ agents in pairwise games of cooperation across a set number of rounds (1 million). In each round, every agent selects another agent to play against. During every interaction, an agent plays their pure strategy independently of the other agent's strategy, and receives a payoff accordingly. Next, the agent who initiated the interaction reinforces their outgoing network connection by the payoff received. An agent can be a player in a maximum of $N$ interaction per round (1 as visitor, and $N-1$ as host). At the end of each round, every agent independently and simultaneously considers imitating another agent with a set probability proportional to the imitation rate ($\lambda$). If an agent is selected to consider imitation, then they randomly select another agent with a probability proportional to the weight of their outgoing and incoming links with other agents. This represents the overall frequency of interaction with each other agent so individuals are more likely to consider agents with which they have more interactions. The selected agent's strategy is imitated if and only if that agent received a strictly greater average payoff per interaction than the imitating agent. We also include an error chance ($e = 0.01$) when agents consider imitation, in which a random strategy is chosen rather than imitating another agent.

Each agent $i$ has a pure strategy ('Social', 'Spiteful') and vector representing their reinforcement weights for the choice of which player to visit: $(w_{i1}, w_{i2}, \dots, w_{in})$ where $w_{ij}$ represents the weight related to player $i$ visiting player $j$. Self-visits are not allowed ($w_{ii} = 0$). At the start of each simulation, half of the agents are 'Social' and half of the agents are 'Spiteful' (results are robust to different starting proportions; Fig. S2). Initial network partner weights are set uniformly to $L/(N-1)$, where $L$ is a parameter determining initial learning weights. We use the convention $L = 9$, so that network partner weights start at $w_{ij} = 1$ for the smallest population we study ($N = 10$). To ensure similar reinforcement learning speed relative to total initial weights across different population sizes, we kept $L$ constant over all simulations. Changing $L$ varies network learning speed because larger $L$ values reduce the relative size of each payoff in comparison to the initial uniform network weights.

The model has two parameters that affect the reinforcement of network weights: discounting ($\delta$) and error ($\epsilon$). Discounting reduces past learning weights as more reinforcement occurs, gradually allowing agents to forget old network connections. Errors represent mistakes, noise, or mutations where an agent selects an interaction partner at random rather than according to their network weights. Both components have been shown to impact the stability and long-run behavior of reinforcement learning.

When selecting an interaction partner for a given round, the probability of choosing agent $j$ is proportional to the current network weights:

$$\Pr(j) = (1-\epsilon)\frac{w_{ij}}{\sum_k w_{ik}} + \epsilon\frac{1}{|N|}, \qquad (1)$$

where $\epsilon$ is the error rate, $N$ is the set of agents, $j \in N$, and $k \in N$.

After each round of interactions, link weights for the selected outgoing links are updated by discounting the prior weight by a factor ($\delta$) and adding the received payoff ($\pi$):

$$w'_{ij} = (1-\delta)w_{ij} + R\pi_i, \qquad (2)$$

where $w'_{ij}$ is the link weight after updating and $\pi_i$ is sum of $i$'s the most recent payoffs. $R$ is the rate of network learning, set to $R = 1$ by default, higher values result in faster learning (results are robust to both small and large learning rates; (Fig. S3). All link weights are updated simultaneously at the end of each round, reflecting the outcome of every interaction an agent was a part of during the round.

After link weights have been updated at the end of a round, every agent independently considers imitating another agent's strategy proportional to the imitation rate ($\gamma$). If an agent is randomly selected to consider imitation, they select a possible imitation partner proportional to their outgoing and incoming link weights associated with every other agent:

$$\Pr(j) = \frac{w_{ij} + w_{ji}}{\sum_k (w_{ik} + w_{ki})}. \qquad (3)$$

If the selected imitation partner has received a higher average payoff over all interactions in the previous round, then the agent considering imitation will adopt the selected agent's strategy.

We measure of correlated interactions for each agent ($a_i$) as a function of the total proportion of the incoming and outgoing link weights of that agent for all connected agents of the same strategy type. Let $s_i$ denote the strategy type of agent $i$, and let $Same_i$ denote the set of agents $j$ that have the same strategy as $i$ ($s_i = s_j$). We calculate the difference between the proportion of same-strategy interactions and the proportion of same-strategies assuming random interaction:

$$a_i = \frac{\sum_{j\in Same_i}(w_{ij} + w_{ji})}{\sum_k (w_{ik} + w_{ki})} - \frac{|Same_i|}{|N|}. \qquad (4)$$

If $a_i > 0$, $i$ interacts with agents of the same type more than would in random interaction and thus are positively correlated, and if $a_i < 0$, they are anti-correlated. This allows us to represent correlated and anti-correlated interaction simultaneously in different sub-populations. The degree of correlated interactions for 'Social' agents is the mean measure of correlated interaction of all agents using the 'Social' strategy; the degree for 'Spiteful' agents is the mean measure for all

## Table 1 Parameters and baseline values.

| Symbol | Parameter description | Values |
|---|---|---|
| $N$ | Population size | 10, 20, 30, 40, 50 |
| $\lambda$ | Strategy imitation rate | 0.01 |
| $e$ | Imitation error chance | 0.01 |
| $w_{ij}^0$ | Initial link weights | $9/(N-1)$ |
| $\delta$ | Link weight discount factor | 0.001, 0.01 |
| $\epsilon$ | Partner choice error rate | 0.01 |
| $R$ | Rate of network learning | 0.01, 0.1, 1, 10 |
| $T$ | Timesteps | $[0 - 1e^6]$ |
| $b$ | Spite–Social payoff | $\frac{10}{11}, \frac{8}{9}, \frac{6}{7}, \frac{4}{5}, \frac{2}{3}$ |
| $c$ | Social–Spite payoff | $\frac{1}{11}, \frac{1}{9}, \frac{1}{7}, \frac{1}{5}, \frac{1}{3}$ |

agents using the 'Spiteful' strategy. The overall measure of correlated interaction for the population ($\bar{a}$) is the mean degree of correlation for all agents in the population regardless of strategy type. Note that in an unstructured population with uniform correlated interaction rates, the classic inequality $r > c/b$ corresponds to the $\bar{a}$-value of $-c/b$ with respect to the stability and invasion conditions for 'Spiteful' behavior (see SI). Also note that the measure of correlated interaction for extinct strategy types is not defined because it is the mean of an empty set, but we plot these values as 0 as these types can be re-introduced by imitation error.

All model parameters are summarized in Table 1.

**Analytic model.** A simplified model enables the derivation of mathematical results that illustrate and further support the key insights of our study. This simplified model uses discrete network links formed by a best-response rule, strategy updating occurs by imitation similarly to the central model. Analytic results from the simplified model further support the simulation results of the central model: endogenous partner choice in dynamic networks allows spite to spread via imitation.

Suppose there are $N$ agents, each with a strategy $s_i \in \{$'Spiteful', 'Social'$\}$. Each agent $i$ at each time $t$ has exactly one outgoing link to one other agent $j \neq i$, represented as $l_{ij}^t \in N \times N$ (a temporal adjacency matrix), incoming links are limited only by the number of agents. At a given time $t$, the interaction set $I_i^t$ for an agent $i$ is the set of all their interaction partners: $I_i^t = \{j | l_{ij}^t \text{ or } l_{ji}^t\}$. Let $u(s_i, s_j)$ be the payoff of $i$'s strategy played against $j$'s strategy.

At the beginning of each time-step, agents select their outgoing link using a myopic best-response rule: link to another agent such that payoffs are maximized if all strategies remain constant. Precisely, each agent forms one link $l_{ij}^t$ for time-step $t$ by choosing randomly (with a uniform distribution) from the set of optimal links $L_i$:

$$L_i = \{l_{ij} | u(s_i, s_j^t) \geq u(s_i^t, s_k^t) \text{ for all } k \in N\}. \tag{5}$$

After links are formed, agents play the game in Fig. 1 with every agent in their interaction set. An agent's average payoff per interaction at time $t$ is:

$$\bar{U}_i^t = \sum_{j \in I_i^t} u(s_i^t, s_j^t)/|I_i^t|. \tag{6}$$

After interacting, agents update their strategies by imitation. Let $B_i^t$ represent the equal or better-performing agents in $i$'s interaction set (including $i$):

$$B_i^t = \{j \in \{I_i^t \cup i\} | \bar{U}_j^t \geq \bar{U}_i^t\}. \tag{7}$$

During imitation $i$ selects a random $j \in B_i^t$ (with a uniform distribution) and adopts that agent's strategy: $s_i^{t+1} = s_j^t$. After imitation, links are updated as above and the process repeats.

*Conditions for invasion and spread of spite.* Note that because the game is dominance solvable, all agents regardless of their own strategies will form links with agents playing 'Social'. Thus, in any mixed population ($N > x > 0$), 'Spiteful' agents will only have 'Social' interaction partners whereas 'Social' agents may have a mix of partners. Consequently, the proportion of 'Social' interactions for a 'Spiteful' agent will always exceed the proportion of 'Social' interactions for anyone they visit. This difference, when $b > c$, allows for the possibility of spite to spread through the population. Given endogenous partner choice, whether spite is expected to spread through imitation depends on the relevant mean payoffs of the agents interacting with 'Spiteful' individuals. To determine this condition, suppose there is a single 'Spiteful' agent $i$. This agent will visit a single 'Social' agent $j$, and not be visited by anyone. Thus, $\bar{U}_i^t = b$.

The 'Social' agent $j$ will be visited by $i$ ($l_{ij}^t$), will visit some other 'Social' agent $k$ ($l_{jk}^t$), and will be visited by $z$ other 'Social' agents (where $0 \leq z \leq N - 2$). Thus,

$\bar{U}_j^t = (c + 1 + z)/(z + 2)$ and $j$ has a chance to imitate $i$ if and only if

$$b \geq \frac{z + 1 + c}{z + 2}. \tag{8}$$

If the above inequality is strict, $i$ will never imitate $j$ and $j$ will eventually imitate $i$'s 'Spiteful' strategy.

Since $c < 1$, there will be some $c$, $b$, and $z$ such that 'Spiteful' will spread by imitation. To determine at what payoff values we can expect spite to begin to spread from a single 'Spiteful' agent, note that all agents form a single link. Hence, the expected value of $z$ is 1 and the expected mean payoff for $j$ is $\text{Exp}(\bar{U}_j^t) = (c + 2)/3$. Thus, spite will be expected to spread by imitation whenever $b > (c + 2)/3$ or:

$$3b - c > 2. \tag{9}$$

Generalizing this, if there are $x$ 'Social' agents and $y$ 'Spiteful' agents, a 'Social' agent $j$ who is interacting with at least one 'Spiteful' individual is expected to be visited by $(y - 1)/x$ other 'Spiteful' individuals and 1 other 'Social' individual. Thus, the expected mean payoff for a 'Social' agent $j$ who is interacting with at least one 'Spiteful' agent is

$$\text{Exp}(\bar{U}_j^t) = \frac{2 + c\left(1 + \frac{y-1}{x}\right)}{3 + \frac{y-1}{x}}. \tag{10}$$

Agent $j$ is expected to imitate a 'Spiteful' individual whenever $\text{Exp}(\bar{U}_j^t) < b$ or

$$b\left(3 + \frac{y-1}{x}\right) - c\left(1 + \frac{y-1}{x}\right) > 2. \tag{11}$$

Comparing the generalized inequality to the case where $y = 1$ reveals that the chance of imitating 'Spiteful' becomes strictly greater the more 'Spiteful' individuals are present in the population. Once a single 'Spiteful' individual begins to spread, it is expected that they will continue to spread through the entire population.

*Measure of correlated interaction during invasion.* We can employ our measure of correlated interaction to this analytic model as well. Suppose there is a single 'Spiteful' individual $i$ in a population of size $N$, and that the condition for the spread of spite is met ($3b - c > 2$). After network links are formed, we can calculate the expected $a$-value of the population. Note that $a_i = -1/N$ for the single 'Spiteful' individual and $a_j = 2/3 - (N-1)/N$ (expected) for the $j$ that $i$ visits, and $a_k = 1 - (N-1)/N$ (expected) for the remaining $N - 2$ individuals. The aggregate correlated interaction is then

$$\bar{a} = (1/N)\left(a_i + a_j + \sum_k a_k\right), \tag{12}$$

which reduces to

$$\bar{a} = \frac{1}{N}(2/3 - 2/N). \tag{13}$$

Using this equation we can see that $\bar{a} > 0$ whenever $N > 3$. Therefore, when spite begins to invade and spread in a population of 4 or more, strategies are (on average) positively correlated. This finding shows clearly that the general conditions derived in classic models (e.g., $r > c/b$) do not generalize to dynamic networks. Indeed, even the very weak condition that $\bar{a} < 0$ is not necessary for spite to invade and spread. This reinforces the important lesson that traditional population-level statistics do not adequately describe social change in dynamic networks.

**Reporting summary.** Further information on research design is available in the Nature Research Reporting Summary linked to this article.

## Data availability
All figures and data can be recreated using our code which has been made publicly available. Simulations were written in C++ and run for $10^6$ rounds of play. Data aggregation and network plots were created using Python.

## Code availability
Replication code available on GitHub: https://doi.org/10.5281/zenodo.3962292.

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

## Acknowledgements

We would like to thank Michael Foley who originally created much of the code used in our simulations and provided advice throughout the project.

## Author contributions

Z.F., P.F., R.S., and C.R. conceived the study, Z.F. conducted the experiment, Z.F. analyzed the results. All authors wrote and reviewed the manuscript.

## Competing interests

The authors declare no competing interests.
