## [Peer Review File · Nature Communications]

Summary

This paper explores the emergence of spite – costly behavior that harms others. It runs simulations where agents update both actions (spite or social) and link weights (that capture the probabilities of interactions). The updates are based on two reduced-form heuristics: (1) strategy imitation – an agent adopts the strategy from a partner who receives higher payoffs, and (2) link updating – an agent puts more weights on partnerships that from which she receives higher payoffs. The dynamics of the simulation can lead to the dominance of spite strategy in population.

I think this is a weak paper on an interesting topic. The key logic behind the results is not clearly presented, and appears to be driven by unjustified assumptions (more in comments #1 and #2). It is not clear whether the results are robust to variations in the detailed assumptions or the parameters of the simulation. The organization and exposition of the paper makes it difficult to read. These all make it hard to identify the important real-world scenarios where this model would fit best and the paper's general insights.

Comments

1. (p.6 last part) One key parameter of the paper, δ , does not appear in any of the rules (equations) of updating. It is very confusing.

I tend to believe there is a typo in the last equation of the page:

$$w_{ij} = (1 - \delta)w_{ij} + R\pi_{ij},$$

which should be

$$w_{ij} = (1 - \delta)w_{ij} + R\pi_{ij}.$$

The corrected equation poses another question: What is the fundamental difference between R (learning rate) and δ (discounting)? It appears that only the ratio $(1 - \delta)/R$ matters for updating, and hence a decrease in R shall have an similar effect as a decrease in δ . However, very different patterns are reported in Fig. 3A and 3C (p.4) – is it because of the specific parameters used in the simulations? The paper does not provide any details to address this confusion.

2. One important part of the simulation is that spite agents get higher payoffs than social agents, and hence are favored in imitation. The paper does not clarify why this occurs. In particular, it is not guaranteed from the interaction pattern that both types target the social agents.

Here is my understanding. Let the total weight of social agents be r_{sp} in a spite agent's neighborhood, and r_{so} in a social agent's neighborhood. Then their payoffs are $\pi_{sp} = r_{sp}b + (1 - r)0$, and $\pi_{so} = r_{so}1 + (1 - r)c$. A necessary condition for $\pi_{sp} > \pi_{so}$ is $r_{sp} > r_{so}$, that is, the spite agents target the social agents faster.

The above underlined part is a hidden and unjustified assumption of the paper. In the current simulation, it is driven by the seemingly innocent normalization $b + c = 1$ (p.2, Fig 1, notes). To see this, consider the network weight updating rule (p.6): for any given constant, say w , we have $\frac{w \pm b}{w} > \frac{w \pm 1}{w} - \frac{w \pm c \pm b}{w \pm c}$. So the spite agents do "learn" relatively faster. When $b + c < 1$, the above can be opposite.

The author(s) may want to clarify the related key assumptions, and identify important applications in which the assumptions would fit. It is also useful to discuss what happens when these assumptions (parameters) do not hold.

3. (p.1 Section 1, para 2) The summary of the emergence of spite due to anti-correlation appears confusing and logically flawed. Firstly, it fails to provide any details (calculations) to justify the results. Secondly, it assumes that when one switches from one action to the other, all her opponents also switch their actions according to the "anti-correlation". This violates the basic logic of game theory: when any agent chooses her action, the actions of all others shall be treated as fixed (at least for the same period). As a side note, given the issue in this part, it is not clear to me whether $-c/b$ shall be used as a benchmark when reporting the results (e.g. p.3, Fig 2, last line).
4. (p.5 Section 3, para 3) The paper claims that the literature "poses a puzzle of how punishment behavior could have emerged in the first place". I do not see the paper addresses that fundamental question: this paper (its simulation) *assumes* half of the population chooses "spite" at the starting point.
5. (related to #4) There is a well developed literature on the theory of reciprocity and retaliation. For instance, Falk & Fischbacher (2006), and Fehr & Gächter (2000).
6. (p.5 Section 3, line 7) Typo: "an " shall be "can"
7. (p.7) It could be helpful to summarize the key assumptions of the "Methods" part up front.

References

- Falk, Armin, and Urs Fischbacher. "A theory of reciprocity." *Games and economic behavior* 54, no. 2 (2006): 293-315.
- Fehr, Ernst, and Simon Gächter. "Fairness and retaliation: The economics of reciprocity." *Journal of economic perspectives* 14.3 (2000): 159-181.

Reviewer #2:

Remarks to the Author:

The authors of the submitted manuscript have used an agent-based model based on a payoff structure called the Prisoner's Delight to show how spite can persist in a dynamically evolving interaction network. This is an interesting paper, and the modelling approach in particular should be of value to other theoreticians. The implications will also be of value to psychologists and evolutionary biologists. As such, I think that this manuscript could be relevant to readers of Nature Communications.

As someone who is not a modeller, I cannot comment on the methods used or the conclusions drawn from the model. I take these at face value. I think readers like me would benefit from having a bit more explicit clarification of the terminology and the reasons for the decisions made in choosing the model. In particular, I was curious about why a Prisoner's Delight game was used. How would this model differ from the more widely used Prisoner's Dilemma and how does this relate to classical approaches such as tit-for-tat (Axelrod & Hamilton, 1981). A bit of background on models that are familiar to behavioural scientists would help put this study into context.

One important theoretical aspect that I think can be improved is the level of explanation (Mayr 1961). This is an important point given that the authors make reference to spite on an evolutionary scale and then refer to papers dealing with proximate behaviours. West, Griffin and Gardner (2007) make this point quite well (see also Gardner & West, 2006). While it was good to see both evolutionary papers and psychological ones referenced, it would help to keep these separate. It might also be better to refer to psychological spite or spiteful behaviour when discussing behaviours and motivations, and biological or functional spite when referring to evolutionary processes (see Jensen, 2010; see also Bshary & Bergmüller, 2008). As it is, starting a paragraph with spite (or counter-selected behaviour, as Hamilton first called it), then moving onto "irrational" at the end requires a fair amount of mental gymnastics on the reader's part.

Furthermore, it would help a non-specialist to see how spite in the model relates to punishment. Punishment looks like something one would find in a standard Prisoner's Dilemma, where tit-for-tit is a retaliatory but forgiving strategy. Does spite differ from punishment? This point becomes particularly relevant for the final paragraph of the Discussion which is focussed on punishment. Punishment did not come up earlier in the paper, even though reinforcement learning is mentioned (with its implications of positive and negative punishment as well as positive and negative reinforcement), therefore the role of spite in punishment is unclear. Why would spite even be needed to explain the model of standard reinforcement learning could work? (As for punishment, it would be helpful to cite a classic paper on the topic: Clutton-Brock & Parker, 1995).

Some minor theoretical and terminological points: A bit more background on the terminology, notably correlation or anti-correlation of strategies, particularly as these relate to negative selection, would be helpful. Another terminological point that could be corrected is the use of the term social to mean prosocial. Social interactions can refer to both positive and negative interactions. Prosocial as a term comes with its own problems, but in the context of this paper, it would be more specific than just social. (Positive social behaviours would likely be the most informative term. Theoretically, it is not obvious why the self-domestication hypothesis would be used to support an argument about the evolution of spite. Boyd and Richerson's accounts would seem more relevant. Given the scope of the paper, it would also be worth mentioning spite in chimpanzees when referring to spiteful behaviour in non-human animals (Jensen, Hare, Call & Tomasello, 2006; Jensen, Call & Tomasello, 2007). Again, this sort of behaviour would need to be treated as distinct from the type of spite biologists refer to in *Wolbachia* and parasitoid wasps. On the topic of mechanisms, more could be said about imitation. There are various models for this, and these could very well influence the transmission of strategies (Hopit & Laland, 2013).

To summarise, I think this is a very interesting and very well-presented paper. It sets itself up against other models for the evolution of spite (e.g., Hamiltonian and Wilsonian negatively-selected spite). I

think by making stronger links between this model and other fitness-based models will help explain the novel benefits of the approach used here. Care needs to be taken to help readers from confusing proximate and ultimate approaches to understanding the emergence of spite.

References

- Axelrod, R. & W. D. Hamilton (1981). "The evolution of cooperation." *Science* 211(4489): 1390-1396.
- Bshary, R. & R. Bergmüller (2008). "Distinguishing four fundamental approaches to the evolution of helping." *Journal of Evolutionary Biology* 21: 405-420.
- Clutton-Brock, T. H. & G. A. Parker (1995). "Punishment in animal societies." *Nature* 373: 209-216.
- Gardner, A. & West, S. A. (2006). "Spite." *Current Biology* 16(17): R662-R664.
- Hoppitt, W. & K. N. Laland (2013). *Social Learning: An Introduction to Mechanisms, Methods, and Models*. Princeton University Press: 62-104.
- Jensen, K., Call, J. & Tomasello, M. (2007). "Chimpanzees are vengeful but not spiteful." *Proceedings of the National Academy of Sciences* 104: 13046-13050.
- Jensen, K. (2010). "Punishment and spite, the dark side of cooperation." *Philosophical Transactions of the Royal Society B: Biological Sciences* 365: 2635-2650.
- Jensen, K., Hare, B., Call, J. & Tomasello, M. (2006). "What's in it for me? Self-regard precludes altruism and spite in chimpanzees." *Proceedings of the Royal Society B-Biological Sciences* 273(1589): 1013-1021.
- Mayr, E. (1961). "Cause and effect in biology." *Science* 134: 1501-1506.
- West, S. A., Griffin, A. S. & Gardner, A. (2007). "Social semantics: Altruism, cooperation, mutualism, strong reciprocity and group selection." *Journal of Evolutionary Biology* 20: 415-432.

Reviewer #3:

Remarks to the Author:

see the attached.

COMMENTS

The spite take a cost to make an even worse harm to others. It is similar to the punishment in the study of evolution of cooperation, which is often addressed as the "second order cooperation dilemma". Thus spite is an evolutionary puzzle, and calls for explanation within evolutionary theory. The authors correctly review the "negative assortment or negative relatedness" in promoting the spiting behavior. And they propose the interplay between network dynamics and spite behavior as an alternative mechanism to promote spiting behavior. Yet, the authors also admit that the dynamical network paves the way to the emergence of the anti-correlation, which was found to be a promoter to the spiting behavior.

On the one hand, I like the manuscript, since it provides an alternative way for the emergence of spite behavior, and in contrast with negative relatedness, the dynamical network provides an dynamical mechanism, which does not explicitly assume relatedness. On the other hand, I find that the findings of the manuscript can be seen as a re-discovery of the cooperation on dynamical networks. In fact, if we rephrase the spite behavior (in this manuscript) as cooperation in prisoners' dilemma (PD), and the social behavior (in this manuscript) as defection in PD game. The basic payoff matrix (in Fig. 1 (2)) is a dominant game, as a PD game is. Thus it is not hard to imagine the results in this manuscript holds, since it is well known in the study of cooperation that dynamical networks promote cooperation both theoretically and empirically. Therefore, I recommend a major revision. The authors should clarify the significance of the work compared with i) PD game on the dynamical networks, as well as ii) other mechanisms to promote the emergence of spite behavior.

1. MAJOR COMMENTS

- The link reinforcement learning is crucial to the model, imho. This includes: "updating their network weight based on the payoff". The principle is "Social agents become correlated with one another and spiteful agents become anti-correlated" based on the link reinforcement learning. Yet, it is not clear to me how robust this principle is to promote the spiting behavior. In other words, I would like to see an alternative model with this principle invariant and leading to at least qualitatively similar results. These alternative models can be shown in the SI.
- The authors show that "This highlights the importance of considering correlation patterns not just at a system level, but also at the level of strategy type". It is better off to cite more relaxant literatures on the cooperation behavior on dynamical networks [to name a few: M.G.Zimmermann and V.M.Eguíluz, Phys.Rev.E 72, 056118 (2005); Wu et.al. PLoS ONE5, e11187(2010)]
- (optional) I totally understand that the work is based on simulation, it would be even better if theoretical investigation can be shown to explain the transient dynamics of spiting behavior (which is rather non-trivial (Fig 3. a,b)). Even a simplified model on some special cases is acceptable.

Changes in this Version

Thank you for inviting us to submit a revised version of our manuscript. We appreciate the comments we received from the three reviewers assigned to our work. As requested, we made several revisions to the manuscript and responded to each review comment point-by-point below. All changes to the manuscript are highlighted using track-changes. The reviewers made excellent constructive suggestions and we have followed their advice. The revised version of the paper includes the following key changes:

1. Revised introduction paragraphs 2 and 3 for clarity and connections to the literature.
2. We expanded the literature review to clarify our contribution and highlight how our work using the Prisoner's Delight relates to work using the Dilemma (introduction, 3rd paragraph). We also included new simulation results of a direct comparison between our model of the Delight with the Dilemma in the Supplement (page 12).
3. Edited and expanded the Methods section for clarity.
4. We added an analytical model (now included in the SI, page 11) which assumes no difference in learning speed and derives the explicit inequalities necessary for the spread of spite in that simplified model.
5. Added a new section in the SI where we derive the $r > c/b$ inequality as well as the explicit connection between this inequality and our measure of endogenous correlated interaction.
6. We added additional simulation results to the Robustness Tests section in the SI to address various concerns raised by the reviewers. In particular, we show that our results are not driven by the normalization $b + c = 1$ (subsection "Payoff Structure", page 15), differences in learning speed (subsection "Network Learning Speed", page 13), and that a single spiteful agent can invade a population (subsection "Starting Proportion", page 13).
7. We revised the narrative through to clarify hard-to-understand sentences (see especially the revised Discussion), be more explicit about our assumptions and modeling parameters (see new Table 1 in the Methods section), and fixed various typos as pointed out by the reviewers.

Response to Reviewer 1

R#1-1. (p.6 last part) One key parameter of the paper, δ , does not appear in any of the rules (equations) of updating. It is very confusing. I tend to believe there is a typo in the last equation of the page:

$$w'_{ij} = (1 - \epsilon)w_{ij} + R\pi_{ij},$$

which should be

$$w'_{ij} = (1 - \delta)w_{ij} + R\pi_{ij}.$$

The corrected equation poses another question: What is the fundamental difference between R (learning rate) and δ (discounting)? It appears that only the ratio $(1-\delta)/R$ matters for updating, and hence a decrease in R shall have an similar effect as a decrease in δ . However, very different patterns are reported in Fig. 3A and 3C (p.4) – is it because of the specific parameters used in the simulations? The paper does not provide any details to address this confusion.

Response: Thank you for identifying the typo in our equation, this has been corrected. In response to your question, changes in learning rate (R) and discounting (δ) can have similar effects at any given round of the simulation, but the effects change over time as learning weights accumulate, which accounts for the differences in results presented. We welcome your probing question and have added a discussion of your insight in the SI (subsection “Network Learning Speed”, page 14). Here is further explanation:

For any given time-step, it is correct that $(1-\delta)$ and (R) behave in similar ways, but because weights accumulate over time while payoffs are constant throughout time, their effects diverge over time. To see this, first note that $(1-\delta)$ is applied to the total accumulated weights (w_{ij}), whereas R is applied only to the payoffs of the current round (π_{ij}). Weights (w_{ij}) will change significantly over the course of the simulation, but payoffs (π_{ij}) are always one of a fixed set of values from the payoff matrix. So early on in the simulation $(1-\delta)$ and (R) have similar effects. However, as the weights increase—during the course of the simulation, as agents accumulate more payoffs w_{ij} becomes very large relative to π_{ij} —the effect of $(1-\delta)$ becomes much larger than the effect of R . To put this another way, δ has a constant proportional effect on the learning behavior in every round, but the effect of R decreases as there are more and more interactions that occur. So, while R impacts early learning significantly, it has less effect once accumulated weights become large. There are values of R that could affect the long run behavior, these would be orders of magnitude larger than those we examine, and such large R values would have the awkward effect of “locking in” the first action to receive any payoff at all.

R#1-2. One important part of the simulation is that spite agents get higher payoffs than social agents, and hence are favored in imitation. The paper does not clarify why this occurs. In particular, it is not guaranteed from the interaction pattern that both types target the social agents.

Here is my understanding. Let the total weight of social agents be r_{sp} in a spite agent's neighborhood, and r_{so} in a social agent's neighborhood. Then their payoffs are $\pi_{sp} = r_{sp}b + (1 - r)0$, and $\pi_{so} = r_{so}1 + (1 - r)c$. A necessary condition for $\pi_{sp} > \pi_{so}$ is $r_{sp} > r_{so}$, that is, the spite agents target the social agents faster.

The above underlined part is a hidden and unjustified assumption of the paper. In the current simulation, it is driven by the seemingly innocent normalization $b + c = 1$ (p.2, Fig 1, notes). To see this, consider the network weight updating rule (p.6): for any given constant, say w , we have

$$\frac{w+b}{w} > \frac{w+1}{w+c} = \frac{w+c+b}{w+c}$$

So the spite agents do “learn” relatively faster. When $b + c < 1$, the above can be opposite.

The author(s) may want to clarify the related key assumptions, and identify important applications in which the assumptions would fit. It is also useful to discuss what happens when these assumptions (parameters) do not hold.

Response: We appreciate the reviewer's thoughtful work here. First, we'd like to note that the reviewer's payoff calculations regarding the requirement that $r_{sp} > r_{so}$ does not imply that spiteful agents target the social agents any faster, it only implies that (at the moment of imitation) social agents must make up a larger proportion of the spiteful agents' total *interactions* than they do for other social agents. This can happen without differences in learning speed because spiteful agents tend to have fewer interaction partners than do social agents. Nevertheless, the reviewer's insight here prompted us to develop a simplified analytic model (now included in the SI, page 12) which assumes no difference in learning speed and derives the explicit inequalities necessary for the spread of spite in that simplified model. The results of this analytic model largely coincide with the simulation results for the full model. We have also made more explicit how the imitation rule operates in the Methods section to add clarity to these points.

We also appreciate the suggestion of looking at payoff values such that $b+c < 1$, which allow for synergistic (non-additive) payoffs for social agents—which presents an interesting variation to explore. In response, we have included additional simulation results investigating this possibility (SI subsection “Payoff Structure”, page 14). The lower $b+c$ becomes, the more difficult it is for spite to evolve, but, provided the ratio of b/c is large enough, spite will still evolve even when $b+c < 1$. The results from the analytic model also suggest that $b+c=1$ is not essential, but rather the ratio of b/c is more important. In summary, your comments have prompted us to strengthen

and expand our work with additional simulations and an analytical model which further enhances our contribution.

R#1-3. (p.1 Section 1, para 2) The summary of the emergence of spite due to anti-correlation appears confusing and logically flawed. Firstly, it fails to provide any details (calculations) to justify the results. Secondly, it assumes that when one switches from one action to the other, all her opponents also switch their actions according to the “anti-correlation”. This violates the basic logic of game theory: when any agent chooses her action, the actions of all others shall be treated as fixed (at least for the same period). As a side note, given the issue in this part, it is not clear to me whether $-c/b$ shall be used as a benchmark when reporting the results (e.g. p.3, Fig 2, last line).

Response: We thank the reviewer for the comment here. We have revised the relevant paragraph for clarity surrounding the explanation of those previous studies results and have also added a section in the SI (Section “Derivation of the classic anti-correlated interaction threshold for spite”, page 11) that shows the derivation of the relevant anti-correlated interaction inequality. The following section “Measure of Endogenous Correlated Interaction” derives the explicit connection with the classic inequality and our measure, and explains that the $-c/b$ value corresponds to the classic inequality with respect to stability and invasion conditions for spite.

To address the reviewers second concern, when an agent in our model changes their strategy via imitation it does not directly cause any other agent to change strategies. For example, if a majority of a social agent’s link weights are connected to other social agents, but the agent imitates a spiteful agent at the end of the round, their degree of correlated interactions will go from positive (correlated with other social) to negative (anti-correlated now that they have switched to spiteful) when it is recalculated in the next round. Lastly, to be clear, our model does not include correlated interaction as a parameter, but rather we only use it as a descriptive statistic for analysis. We hope the short section we have added to the SI clarifies any confusion on this point. We have also made explicit about how the measure is calculated in the Methods.

R#1-4. (p.5 Section 3, para 3) The paper claims that the literature “poses a puzzle of how punishment behavior could have emerged in the first place”. I do not see the paper addresses that fundamental question: this paper (its simulation) assumes half of the population chooses “spite” at the starting point.

Response: The reviewer is correct that many of the results we show had populations initialized with 50% spiteful individuals. We have added simulation results in the SI showing that spite can invade populations initialized with just a single spiteful individual (SI subsection “Starting Proportion”, page 13). We have added slightly more discussion around these results and now mention this robustness test in the main results section. Furthermore, the analytic model we have added in our revision, also derives a similar result (SI section “Analytic Model”, page 12).

The substantive insight from the simulation results and the analytic model is: Yes, a single spiteful agent can invade a population of social agents.

R#1-5. (related to #4) There is a well developed literature on the theory of reciprocity and retaliation. For instance, Falk & Fischbacher (2006), and Fehr & Gächter (2000).

Response: Thanks for drawing our attention to these papers. We have incorporated citations into the discussion and revised the paragraph on punishment for clarity.

R#1-6. (p.5 Section 3, line 7) Typo: “an” shall be “can”.

Response: We fixed this typo, thanks for finding.

R#1-7. (p.7) It could be helpful to summarize the key assumptions of the “Methods” part up front.

Response: Thank you for the suggestion, we have added a table that contains the key parameters and the values used in the simulations to the Methods section (page 8).

In summary, we wish to thank the Reviewer for the careful reading of the manuscript, and for prompting us to address the learning speed issue – we feel that this, and other changes we made in response to the Reviewer’s recommendations, have considerably strengthened the manuscript.

Response to Reviewer 2

R#2-1. The authors of the submitted manuscript have used an agent-based model based on a payoff structure called the Prisoner's Delight to show how spite can persist in a dynamically evolving interaction network. This is an interesting paper, and the modelling approach in particular should be of value to other theoreticians. The implications will also be of value to psychologists and evolutionary biologists. As such, I think that this manuscript could be relevant to readers of Nature Communications.

As someone who is not a modeller, I cannot comment on the methods used or the conclusions drawn from the model. I take these at face value. I think readers like me would benefit from having a bit more explicit clarification of the terminology and the reasons for the decisions made in choosing the model. In particular, I was curious about why a Prisoner's Delight game was used. How would this model differ from the more widely used Prisoner's Dilemma and how does this relate to classical approaches such as tit-for-tat (Axelrod & Hamilton, 1981). A bit of background on models that are familiar to behavioural scientists would help put this study into context.

Response: Thank you for the comment. We have revised two paragraphs in the introduction to make it more clear how our investigation of spite (using the Delight) relates to investigations of altruism (using the Dilemma). Another reviewer also suggested there were connections between our study and those that have examined the Prisoner's Dilemma on dynamic networks, so we have included both references to those previous studies. Furthermore, we added a comparison section in the SI where we show simulation results when we use the Dilemma and how those compare to the main results presented (Section "Comparison to Prisoner's Dilemma", page 13).

R#2-2. One important theoretical aspect that I think can be improved is the level of explanation (Mayr 1961). This is an important point given that the authors make reference to spite on an evolutionary scale and then refer to papers dealing with proximate behaviours. West, Griffin and Gardner (2007) make this point quite well (see also Gardner & West, 2006). While it was good to see both evolutionary papers and psychological ones referenced, it would help to keep these separate. It might also be better to refer to psychological spite or spiteful behaviour when discussing behaviours and motivations, and biological or functional spite when referring to evolutionary processes (see Jensen, 2010; see also Bshary & Bergmüller, 2008). As it is, starting a paragraph with spite (or counter-selected behaviour, as Hamilton first called it), then moving onto "irrational" at the end requires a fair amount of mental gymnastics on the reader's part.

Response: Your comment here is much appreciated. You are right to insist that we be sensitive to both the scale of explanation (proximate vs. ultimate) point as well as the complex relationship between psychological and evolutionary processes, and we should have been clearer about how the formal models apply to biological (more ultimate) versus cultural (more proximate) evolutionary settings. We have significantly revised the discussion section to help clarify some of these issues as well as incorporating the insightful connection to the comparative psychological literature on spite you suggested. See, in particular, paragraphs 2 and 3 of the Discussion.

Regarding the specific use of the ultimate/proximate distinction, given the dual interpretations (biological and cultural) and the complications that have been identified with the distinction (see., e.g., Laland et al 2011), we do not see a clear and uncontroversial way to apply it to our formal modeling work. However, the general point—that we needed to be clearer about what our models show—is well taken and incorporated into the revisions.

Regarding the West, Griffin and Gardner (2007) paper and the Gardner and West (2006) entry in *Current Biology*, we prefer to cite their more recent review article in *Science* (reference 11) for two reasons. First, given the broad readership of *Nature Communications* the review article is more accessible; and second, in the review much of their earlier conceptual work is discussed and cited.

R#2-3. Furthermore, it would help a non-specialist to see how spite in the model relates to punishment. Punishment looks like something one would find in a standard Prisoner's Dilemma, where tit-for-tit is a retaliatory but forgiving strategy. Does spite differ from punishment? This point becomes particularly relevant for the final paragraph of the Discussion which is focussed on punishment. Punishment did not come up earlier in the paper, even though reinforcement learning is mentioned (with its implications of positive and negative punishment as well as positive and negative reinforcement), therefore the role of spite in punishment is unclear. Why would spite even be needed to explain the model of standard reinforcement learning could work? (As for punishment, it would be helpful to cite a classic paper on the topic: Clutton-Brock & Parker, 1995).

Response: Thanks for the comments and questions here—we have realized our original draft was unclear on these points. Just to clarify our approach: following most of the evolutionary literature on punishment we understand it as conditional harmful behavior that is triggered by violation or deviation from some behavioral regularity or norm. For instance, free-riding triggers a costly and harmful punishment from other members in the group. Spiteful behavior is simply costly and harmful; the behavior need not be conditional on anything. If spite evolves in a population then evolution can “tinker” with this costly harmful behavior by making the response conditional. We have significantly revised the discussion on punishment to help make this point

clear and we have added the citation to Clutton-Brock and Parker paper you suggested (see paragraph 4 of the Discussion section).

R#2-4. Some minor theoretical and terminological points: A bit more background on the terminology, notably correlation or anti-correlation of strategies, particularly as these relate to negative selection, would be helpful. Another terminological point that could be corrected is the use of the term social to mean prosocial. Social interactions can refer to both positive and negative interactions. Prosocial as a term comes with its own problems, but in the context of this paper, it would be more specific than just social. (Positive social behaviours would likely be the most informative term. Theoretically, it is not obvious why the self-domestication hypothesis would be used to support an argument about the evolution of spite. Boyd and Richerson's accounts would seem more relevant. Given the scope of the paper, it would also be worth mentioning spite in chimpanzees when referring to spiteful behaviour in non-human animals (Jensen, Hare, Call & Tomasello, 2006; Jensen, Call & Tomasello, 2007). Again, this sort of behaviour would need to be treated as distinct from the type of spite biologists refer to in *Wolbachia* and parasitoid wasps. On the topic of mechanisms, more could be said about imitation. There are various models for this, and these could very well influence the transmission of strategies (Hoppit & Laland, 2013).

Response: Thanks for pushing us to clarify these issues. With respect to terminology, we have made an effort to clarify correlated and anti-correlated interactions, both in the language of the Introduction and Discussion, and in formal respects in the Methods (we now specify exactly how we are measuring it) and the SI (we now explicitly derive relationships between the measure we must use on dynamic networks and traditional Hamiltonian approaches).

With respect to the self-domestication hypothesis, and specifically Wrangham's version of it, we have made an effort to clarify the connection in the Introduction: since our model illustrates how spite (costly harm) can evolve, it shows how Wrangham's feedback loop of coalitionary violence leading to self-domestication could get started (i.e., why harmful behavior would evolve in the first place). We have also addressed Boyd and Richerson's hypothesis in the Discussion.

With respect to the connection to spite in chimps, we welcome the idea and have included these citations in the Discussion.

R#2-5. To summarise, I think this is a very interesting and very well-presented paper. It sets itself up against other models for the evolution of spite (e.g., Hamiltonian and Wilsonian negatively-selected spite). I think by making stronger links between this model and other fitness-based models will help explain the novel benefits of the approach used here. Care needs to be taken

to help readers from confusing proximate and ultimate approaches to understanding the emergence of spite.

Response: We appreciate the reviewers comments and words of encouragement. We have revised the content of both the Introduction and Discussion to more clearly elaborate the links between our model and other fitness-based work. We have also highlighted what makes our work novel and interesting in comparison to previous research. Finally, we have added an explanation on the differences between proximate and ultimate behavior to add context to the results of our simulations which help make the manuscript accessible to a broad readership.

In summary, we wish to thank the Reviewer for the thoughtful comments and recommendations that have considerably improved the manuscript.

References

Laland KN, Sterelny K, Odling-Smee FJ, Hoppitt W, Uller T (2011) Cause and effect in biology revisited: is Mayr's proximate–ultimate dichotomy still useful? *Science* 334:1512–1516

Response to Reviewer 3

R#3-1. The spite take a cost to make an even worse harm to others. It is similar to the punishment in the study of evolution of cooperation, which is often addressed as the "second order cooperation dilemma". Thus spite is an evolutionary puzzle, and calls for explanation within evolutionary theory. The authors correctly review the "negative assortment or negative relatedness" in promoting the spiting behavior. And they propose the interplay between network dynamics and spite behavior as an alternative mechanism to promote spiting behavior. Yet, the authors also admit that the dynamical network paves the way to the emergence of the anti-correlation, which was found to be a promoter to the spiting behavior.

On the other hand, I find that the findings of the manuscript can be seen as a re-discovery of the cooperation on dynamical networks. In fact, if we rephrase the spite behavior (in this manuscript) as cooperation in prisoners' dilemma (PD), and the social behavior (in this manuscript) as defection in PD game. The basic payoff matrix (in Fig. 1 (2)) is a dominant game, as a PD game is. Thus it is not hard to imagine the results in this manuscript holds, since it is well known in the study of cooperation that dynamical networks promote cooperation both theoretically and empirically. Therefore, I recommend a major revision. The authors should clarify the significance of the work compared with i) PD game on the dynamical networks, as well as ii) other mechanisms to promote the emergence of spite behavior.

Response: Thank you for the suggestion. We revised paragraphs 2 and 3 in the introduction to clarify the connections between our work and existing work. We also added an entirely new section to the SI (Section "Comparison to Prisoner's Dilemma", page 13), to explain the similarity and differences of our results to the emergence of cooperation in the Dilemma game. In both cases, a similar network dynamic emerges of agents seeking to avoid interactions with spiteful or defecting agents. The key difference is that the resulting strategic evolution is dependent on the kind of imitation rule used. In the Prisoner's Dilemma the dominated strategy—cooperation—emerges only when *total payoff* comparison is used to determine imitation. Conversely, in the Prisoner's Delight the dominated strategy—spite—emerges only when *average payoff* comparison is used.

More generally, the connections between the Dilemma and the Delight are fairly complicated. Some scholars, such as Lehmann et al. (2006), argue that spite is a form of indirect altruism (meaning there should be parallels between cooperation in the Dilemma and spite in the Delight). Others argue (e.g., West and Gardner 2009) that apparent spite is usually a form of selfishness, in which case we should expect parallels between defection and spite. Others (e.g., Smead and Forber 2012) maintain it is a distinct type of behavior from either selfishness or altruism. These nuances are now better reflected in the Introduction (paragraphs 2 and 3) as

well as explained in the SI section “Comparison with the Prisoner’s Dilemma”. However, given the broad readership of Nature Communications, we suspect that a much longer and more detailed discussion that would be necessary to handle these formal nuances about definitions and technical details would not be appropriate.

R#3-2. The link reinforcement learning is crucial to the model, imho. This includes: “up-dating their network weight based on the payoff”. The principle is “Social agents become correlated with one another and spiteful agents become anti-correlated” based on the link reinforcement learning. Yet, it is not clear to me how robust this principle is to promote the spiting behavior. In other words, I would like to see an alternative model with this principle invariant and leading to at least qualitatively similar results. These alternative models can be shown in the SI.

Response: We appreciate the interesting suggestion and have added an alternative analytic model to the SI (page 12). The alternative model does not rely on reinforcement learning and uses instead a best-response rule for establishing network links (and thus for generating anti-correlated interaction), and the results are largely similar to those seen in the simulations of the primary model.

R#3-3. The authors show that “This highlights the importance of considering correlation patterns not just at a system level, but also at the level of strategy type”. It is better off to cite more relaxant literatures on the cooperation behavior on dynamical networks [to name a few: M.G.Zimmermann and V.M.Egu?luz, Phys.Rev.E 72, 056118 (2005); Wu et.al. PLoS ONE5, e11187(2010)]

Response: Thanks for the suggestion. We have re-written part of the introduction (paragraphs 2 and 3) and added the suggestion citations to clarify both the previous contributions in this area as well as how our study relates to these.

R#3-4. (optional) I totally understand that the work is based on simulation, it would be even better if theoretical investigation can be shown to explain the transient dynamics of spiting behavior (which is rather non-trival (Fig 3. a,b)). Even a simplified model on some special cases is acceptable.

Response: We agree and although the primary model is somewhat intractable analytically, a simplified version would be helpful. Based on your comment we have added an analytic model to our paper (“Analytic Model” in the SI, page 12). This model uses dynamics that are simple enough to derive analytic results. The results of this analytic model largely coincide with the simulation results for the full model. These analytic results provide some theoretical insight into the central features for why spite spreads in dynamic networks. We think this addition improves

the paper and we thank the reviewer for the suggestion here.

In summary, we wish to thank the Reviewer for the careful reading of the manuscript, and for prompting us to add an analytic model that does not rely on reinforcement learning. We feel that this, and other changes we made in response to the Reviewer's recommendations, have considerably strengthened the manuscript.

REFERENCES

Lehmann, L., Bargum, K. and Reuter, M. An evolutionary analysis of the relationship between spite and altruism. *J. Evol Bio* 19, 1507--1516 (2006).

West, S. A. and Gardner, A. Altruism, spite, and greenbeards. *Science* 327: 1341--1344 (2010).

Smead, R. and Forber, P. The evolutionary dynamics of spite in finite populations. *Evolution* 67: 698-707 (2012).

Reviewer #1:

This revision has significantly improved the clarity and quality of the paper. There are still many typos and places that can be improved.

Comments

1. On the analytic model (related to previous comment R1-2). I do appreciate the introduction of the analytic model. It is very stylized but helps to clarify the main mechanism of the paper: it justifies $r_{sp} > r_{so}$, that is, *social agents make up a larger proportion of the spiteful agents' interactions than they do for other social agents*. In particular, everyone prefers to interact with the social type, and therefore spiteful types rarely gain (new) links. Consequently, social agents' interactions will have some incoming links from spiteful agents, while spiteful agents can eventually get rid of such "harmful" links – as no one wants to interact with the spiteful! This is made possible by endogenizing the choices of partners.

This message has not been emphasized enough by the current draft. The above italicised sentence appeared in the response letter, but not the paper. The author(s) shall state these important insights explicitly and clearly in the analytic model.

To make the point even more powerful, the author(s) shall consider compare the current results to a benchmark case with fixed partners/interactions. I provide one such example here: consider the following a ring network, which is an extreme case of initial interaction patterns described in the analytic example (each agent gets 1 outgoing link and 1 incoming link).

$t = 1$: suppose one agent, say 1, is spiteful. All others are social. Easy to see 2 and 3 will imitate 1 when $2b - c > 1$.

$t = 2$: now 1, 2 and 3 are spiteful. Will 4/5 imitate 2/3? Never!! Since $U_2^2 = (0 + b)/2$, always smaller than $U_4^2 = (c + 1)/2$.

This simple example shows that imitation only is NOT enough for spiteful behaviors to spread to the entire population; actually, spiteful behaviors do spread (from 1 to

2/3), but stop spreading onwards. (This point can be contrasted to the observation that with partner selections, more spiteful agents always make the spread more easily). In sum, I found the analytic model helpful, but the current exposition has not clearly emphasized the key insights. In addition, I would suggest to move the (revised) analytic model to the main text.

2. (Related to previous comment R1-3) I appreciate the derivation of the classic anti-correlated interaction part, and it is the way I understood when reading the initial submission. However, I still find the logic flawed. As the author(s) write in p.12:

$$F(\textit{Social}, x) = (1 - r)(x + (1 - x)c) + rc$$

$$F(\textit{Spite}, x) = (1 - r)bx + rb$$

While $(1 - r)$ of opponents remain random, the fraction of r *change their actions* according to the action of the agent of consideration: when he plays Social, the r -opponents play Spiteful; while when he plays Spiteful, the r -opponents play Social.

Again, the above violates the basic assumption we often use to analyze games (e.g. in a Nash equilibrium): the opponents' actions shall be viewed as given/fixed, and should not alter as one's own action changes.

The author(s)' response in the letter fails to not address this concern.

Minor points

3. I personally find the emphasis of “anti-correlation” and “correlation” a bit misleading. The focus on these terms (esp. in the abstract and discussions) make it harder to see the paper's insights regarding the co-evolution of behaviors and interaction patterns. But my judgement could be wrong due to my limited understanding to the related literature.
4. p.8 $w'_{ij} = (1 - \varepsilon\delta)w_{ij} + R\pi_{ij}$ or $w'_{ij} = (1 - \delta)w_{ij} + R\pi_{ij}$?
5. The exposition still appears a bit sloppy overall. There are several typos. For instance, several citations do not display properly (as ?) in text (p.6).
6. p.13 Definition of \bar{U}_i^t : $j \in L_i$ should be $j \in I_i^t$?
7. p.13 before the equation $b > \frac{z+1+c}{z+2}$ “j will imitate i if and only if” is not precise. By the imitation rule introduced in this analytic model, there is some probability that j imitates i when $b = \frac{z+1+c}{z+2}$.

Reviewer #2:

Remarks to the Author:

The authors of the submitted manuscript on the Prisoner's Delight have done an excellent job addressing this reviewer's comments. I find the theoretical aspects easier to follow, and importantly, the relevance to other researchers is clearer. This paper will be of value for people studying the evolution of spite, and sociality more generally.

There are only two minor suggestions that I can make. First, and most importantly, the authors state in the introduction: "To illustrate this, consider a population where individuals are of two types Social or Spiteful." It would help if the authors were more explicit about what "Social" means here. My assumption is that it means correlated, in that the agents interact with other "altruistic" agents. If so, I wonder whether it is better to refer to these agents as "altruists", consistent with other literature. Altruistic (or pro-social) is the opposite of spiteful; social has a much broader range of meanings. Alternatively, if "Social" means only that these individuals associate with non-spiteful agents (i.e., shunning them), then perhaps this term will work. The authors might need to help readers like me to avoid confusion.

Second, I wish I could help the authors with the citations for the following: "Punishment can influence social interaction in many possible ways, stabilizing cooperation , (or any other behavior 52, or through the interaction of reciprocity and retaliation ??." I am not entirely clear what the authors mean here, so I cannot think of which literature they are referring to.

While I cannot comment on the model or critique its novelty vis a vis other theoretical papers based on Hamilton's cost-benefit model, I found the paper very interesting and potentially important for discussions about the evolution of spite and punishment.^[1]_[SEP]

Reviewer #3:

Remarks to the Author:

see the attached

COMMENTS

The authors have replies most of my questions. I appreciate the efforts authors devote to propose an analytical model. Yet, I still have some concerns, which I would be happy to see the revisions.

1. MAJOR COMMENTS

- It is nice to have a paragraph in the revised version to discuss the similarities and differences between PD game and spite game. The authors claim that "The key difference is that the resulting strategic evolution is dependent on the kind of imitation rule used. In the Prisoner's Dilemma the dominated strategy^a cooperation^a emerges **only** when ?total payoff comparison is used to determine imitation. Conversely, in the Prisoner's Delight the dominated strategy^a spite^a emerges **only** when ?average payoff is used", but why? I guess it has to be some intuitive reasoning behind the two games.

Changes in this Version

Thank you for inviting us to submit a revised version of our manuscript. We appreciate the comments we received from the three reviewers assigned to our work. As requested, we moved the *Analytic Model* from the *SI* to the *Methods* Section, and we made several minor revisions to the manuscript and responded to each review comment point-by-point below. All changes to the manuscript are highlighted using track-changes. The reviewers made excellent constructive suggestions and we have followed their advice. The revised version of the paper includes the following key changes:

1. We moved the *Analytic Model* from the *SI* to the *Methods* Section of the main text.
2. We added to the introduction to better clarify the meaning of the *Social* and *Spiteful* strategy types. We also carefully revised our writing to ensure we consistently use “*Spiteful*” and “*Social*” (i.e., capitalized and italics) when we refer to spiteful and social strategies, but use “spite” (lower case and not italicized) when we refer to spiteful behavior generally and not the specific strategy. We also had the entire paper proof-read by a copy editor to address any remaining mistakes and improve the clarity of some hard-to-understand sentences.
3. To better clarify one of the key insights of the simulation and analytical model, we added several sentences throughout the main text and the *Supplementary Information* to emphasize that by endogenizing the choice of partners, social agents evolve to make up a larger proportion of the spiteful agents’ interactions than they do for social agents.
4. We also added a short discussion to the Supplement that further elaborates on the benefits and contribution of our approach of analyzing the co-evolution of networks and strategies instead of relying on a global correlation.
5. We extended the discussion in the *Supplementary Information* comparing the outcome of our model in the prisoner’s delight to the prisoner’s dilemma to better explain why the non-dominant strategy emerges under average and total comparison respectively.
6. Finally, to emphasize the importance of having the coevolution of network and strategy, we added to the *Supplementary Information* a section (“The Importance of Coevolution of Strategy and Network”) that includes the special case of a network with fixed structure suggested by Reviewer 1 as an example.

Response to Reviewer 1

R#1-1. On the analytic model (related to previous comment R1-2). I do appreciate the introduction of the analytic model. It is very stylized but helps to clarify the main mechanism of the paper: it justifies $r_{sp} > r_{so}$, that is, *social agents make up a larger proportion of the spiteful agents' interactions than they do for other social agents*. In particular, everyone prefers to interact with the social type, and therefore spiteful types rarely gain (new) links. Consequently, social agents' interactions will have some incoming links from spiteful agents, while spiteful agents can eventually get rid of such "harmful" links – as no one wants to interact with the spiteful! This is made possible by endogenizing the choices of partners.

This message has not been emphasized enough by the current draft. The above italicised sentence appeared in the response letter, but not the paper. The author(s) shall state these important insights explicitly and clearly in the analytic model.

To make the point even more powerful, the author(s) shall consider comparing the current results to a benchmark case with fixed partners/interactions. I provide one such example here: consider the following a ring network, which is an extreme case of initial interaction patterns described in the analytic example (each agent gets 1 outgoing link and 1 incoming link).

$t = 1$: suppose one agent, say 1, is spiteful. All others are social. Easy to see 2 and 3 will imitate 1 when $2b - c > 1$.

$t = 2$: now 1, 2 and 3 are spiteful. Will 4/5 imitate 2/3? Never!! Since $U_2^2 = (0 + b)/2$, always smaller than $U_4^2 = (c + 1)/2$.

This simple example shows that imitation only is NOT enough for spiteful behaviors to spread to the entire population; actually, spiteful behaviors do spread (from 1 to 1 2/3), but stop spreading onwards. (This point can be contrasted to the observation that with partner selections, more spiteful agents always make the spread more easily).

In sum, I found the analytic model helpful, but the current exposition has not clearly emphasized the key insights. In addition, I would suggest to move the (revised) analytic model to the main text.

Response: We are very happy to hear that you liked the analytical model. As you suggested, we have moved the analytic model to the *Methods* section of the main text. In so doing we also did a careful check for any typos, made some minor revisions for readability and to add emphasis on some of the key insights. We also made the following minor changes to integrate the presentation of our results better. First, following your suggestion, we added a few sentences to the results section explaining why social agents make up a larger proportion of the spiteful agents' interactions than they do for other social agents. Second, we added a section to the SI that includes a discussion of the special case of our analytic model that you suggested along with a discussion of the implications of this example. This will help to emphasize to readers of the importance of the coevolution of network and strategy, particularly that the endogenous partner choice is essential to the spread of spite. Thank you for this great suggestion. We also added an explicit mention of the proportional interaction point you raised in the analytic model. Finally, we revised the results section to draw more attention to the results of the analytic model and present it more prominently.

R#1-2. (Related to previous comment R1-3) I appreciate the derivation of the classic anticorrelated interaction part, and it is the way I understood when reading the initial submission. However, I still find the logic flawed. As the author(s) write in p.12:

$$\begin{aligned}F(\text{Social}, x) &= (1 - r)(x + (1 - x)c) + rc \\F(\text{Spite}, x) &= (1 - r)bx + rb\end{aligned}$$

While $(1 - r)$ of opponents remain random, the fraction of r change their actions according to the action of the agent of consideration: when he plays Social, the r -opponents play Spiteful; while when he plays Spiteful, the r -opponents play Social.

Again, the above violates the basic assumption we often use to analyze games (e.g. in a Nash equilibrium): the opponents' actions shall be viewed as given/fixed, and should not alter as one's own action changes.

The author(s)' response in the letter fails to not address this concern.

Response: We appreciate the thoughts on this point and agree with the referee that there are some flaws with this global parameter approach that is often used in the literature. We've added some clarification to the 2nd paragraph of the introduction as well as expanded the discussion of the classic condition in the supplement. Our view is that the classic condition and accompanying framework (the r -parameter) has some significant limitations, the lack of fit with standard game theory being among them. The changes include an articulation of our doubts/reservations over the classical approach, which also motivate the central model in our study (which uses endogenous assortment/correlation rather than using an exogenous parameter).

We would like to stress that our models here do *not* use this exogenous parameter approach, the $r > c/b$ is not *our* result, and that it is used *only* as a point of comparison. We do believe it is an important comparison to include because (1) it is a well-known result in the literature on spite and (2) the results from our model reveal important limitations to the methods used in those previous studies.

R#1-3. I personally find the emphasis of “anti-correlation” and “correlation” a bit misleading. The focus on these terms (esp. in the abstract and discussions) make it harder to see the paper’s insights regarding the co-evolution of behaviors and interaction patterns. But my judgement could be wrong due to my limited understanding to the related literature.

Response: We appreciate the reviewer sharing their thoughts on the framing of our key insights. With our work we hope to contribute to an ongoing discussion in the literature that directly addresses the role of anti-correlated interactions in the emergence of spite. Specifically, we believe our approach to introducing anti-correlation highlights several flaws in the classical correlation approach. We also hope that our work will be of interest to an interdisciplinary audience who may not already be familiar with correlated interactions. We recognize that different disciplines may refer to this phenomena in different ways (biologists often use the term “assortment” for instance), but we feel that using the more abstract “correlated interaction” better captures the breadth of the idea. To better highlight the key insight regarding the necessary coevolution of behaviors and interaction we have implemented the reviewer’s suggestions from their first comment. Furthermore, we have revised our language in the 4th paragraph of the Results section (starting “The endogenous partner choices ...”) to be clearer when we discuss (anti-)correlated behavior and how our results relate to the earlier literature that used an exogenous global correlation parameter. Finally, we also carefully revised our writing to ensure we consistently use “*Spite*” and “*Social*” (i.e., capitalized and italics) when we refer to spite and social strategies, but use “spite” (lower case and not italicized) when we refer to spiteful behavior generally and not the specific strategy. In the track changes version, this visually exaggerates the amount of changes we made

R#1-4. p.8 $w'_{ij} = (1 - \epsilon\delta)w_{ij} + R\pi_{ij}$ or $w'_{ij} = (1 - \delta)w_{ij} + R\pi_{ij}$?

Response: The correct equation is, $w'_{ij} = (1 - \delta)w_{ij} + R\pi_{ij}$. This equation provides the update rule for the directed link connecting the visitor to the selected host. This update rule does not depend on if the host was selected based on link weights or by a random error.

R#1-5. The exposition still appears a bit sloppy overall. There are several typos. For instance, several citations do not display properly (as ?) in text (p.6)

Response: We apologize for the mistakes missed in our previous draft. We have made sure the missing citation errors did not occur in our updated difference tracking document. We have also had several colleagues review our manuscript to identify additional outstanding errors and these have been corrected.

R#1-6. p.13 Definition of $\cup_i^t : j \in L_i$ should be $j \in I_{ij}^t$?

Response: Thank you for identifying the typo in our equation. When determining the average payoff per interaction we should sum over the set of an agent's current incoming and outgoing links, I_i^t , not the set of optimal links. I_{ij}^t is the set of an agent's outgoing links only.

R#1-7. p.13 before the equation $b > (z+1+c)/(z+2)$ "j will imitate i if and only if" is not precise. By the imitation rule introduced in this analytic model, there is some probability that j imitates i when $b = (z+1+c)/(z+2)$.

Response: We thank the reviewer for identifying this mistake, and for their careful reading. We have corrected this passage to say "j has a chance to imitate i if and only if $b \geq (z+1+c)/(z+2)$. If the above inequality is strict, i will never imitate j and j will eventually imitate i's *Spiteful* strategy." In making this correction, we also identified that the definition of the B_i^t set was misleading and potentially problematic. This has been corrected. The revised version is both simpler and better matches the simulation model.

Response to Reviewer 2

R#2-1. The authors of the submitted manuscript on the Prisoner's Delight have done an excellent job addressing this reviewer's comments. I find the theoretical aspects easier to follow, and importantly, the relevance to other researchers is clearer. This paper will be of value for people studying the evolution of spite, and sociality more generally.

There are only two minor suggestions that I can make. First, and most importantly, the authors state in the introduction: "To illustrate this, consider a population where individuals are of two types Social or Spiteful." It would help if the authors were more explicit about what "Social" means here. My assumption is that it means correlated, in that the agents interact with other "altruistic" agents. If so, I wonder whether it is better to refer to these agents as "altruists", consistent with other literature. Altruistic (or pro-social) is the opposite of spiteful; social has a much broader range of meanings. Alternatively, if "Social" means only that these individuals associate with non-spiteful agents (i.e., shunning them), then perhaps this term will work. The authors might need to help readers like me to avoid confusion.

Response: We are glad to hear you think we did a great job with our revisions and we want to thank you for taking another look at our paper. We thank the reviewer for sharing their concerns about possible confusion over the chosen names for our strategy types. We believe these names are consistent with previous literature on similar topics, but we do understand how the *Social* name could be misleading. To ensure readers have better clarity of the implications of each strategy type we added a clarifying sentence right after the strategy types are introduced: "*Spiteful* agents choose to pay a cost to inflict harm on their interaction partner, while *Social* agents do not pay a cost and do not inflict harm (note that ``*Social*'' is used minimally to refer to non-spiteful behavior)." The strategy types only relate to an agent's willingness to inflict harm at a cost, and do not relate to an agent's willingness or desire to interact with any strategy type.

R#2-2. Second, I wish I could help the authors with the citations for the following: "Punishment can influence social interaction in many possible ways, stabilizing cooperation , (or any other behavior 52, or through the interaction of reciprocity and retaliation ??." I am not entirely clear what the authors mean here, so I cannot think of which literature they are referring to.

While I cannot comment on the model or critique its novelty vis a vis other theoretical papers based on Hamilton's cost-benefit model, I found the paper very interesting and potentially important for discussions about the evolution of spite and punishment.

Response: This was an unfortunate mistake on our part. The missing citations were caused by a compiling error in the track-changes version of the manuscript---the actual version of the text was never affected. This has been fixed. The correct missing citations were, *Fairness and retaliation: the economics of reciprocity* (Fehr & Gächter, 2000) and *A theory of reciprocity* (Falk & Fischbacher, 2006). We apologize for any confusion caused by our mistake.

Response to Reviewer 3

R#3-1. It is nice to have a paragraph in the revised version to discuss the similarities and differences between PD game and spite game. The authors claim that "The key difference is that the resulting strategic evolution is dependent on the kind of imitation rule used. In the Prisoner's Dilemma the dominated strategy; a cooperation; a emerges **only** when "total payoff" comparison is used to determine imitation. Conversely, in the Prisoner's Delight the dominated strategy; a spite; a emerges **only** when "average payoff is used", but why? I guess it has to be some intuitive reasoning behind the two games.

Response: We thank the reviewer for their feedback. The intuitive reason for this difference is that the Social/Cooperative agents attract more visitors, have more total interactions, and so get higher total payoffs even if they are at a relative disadvantage in each interaction. In the Dilemma, cooperation is at a relative disadvantage to defection in each interaction, so imitating based on average payoff will favor defection. In the Delight, spite is costly and deters visitors so will decrease total payoff, but in each interaction, it generates a relative advantage over their partners because $b > c$ (the harm done is greater than the cost paid). Imitation based on average payoffs makes agents more sensitive to the **relative** advantages, and hence allows spite to spread. We added the below clarifying sentences to the supplement in the "Comparison to Prisoner's Dilemma" Section: *"The reason for this difference is that spite spreads by generating relative advantages, at an absolute cost, on a per-interaction basis. If agents track total payoff rather than average payoff, they become less sensitive to these relative advantages and imitation become more sensitive to the total number of interactions. Both agents playing Cooperate in the Dilemma and Social agents in the Delight attract more visitors and have more interactions, and consequently have higher total payoffs."*

Reviewers' Comments:

Reviewer #1:

Remarks to the Author:

The revised manuscript has addressed the comments I raised in my previous report.